# The Expression of Follistatin-like 1 Protein Is Associated with the Activation of the EMT Program in Sjögren’s Syndrome

**DOI:** 10.3390/jcm11185368

**Published:** 2022-09-13

**Authors:** Margherita Sisto, Domenico Ribatti, Giuseppe Ingravallo, Sabrina Lisi

**Affiliations:** 1Department of Basic Medical Sciences, Neurosciences and Sensory Organs (SMBNOS), Section of Human Anatomy and Histology, University of Bari “Aldo Moro”, 70121 Bari, Italy; 2Department of Emergency and Organ Transplantation (DETO), Pathology Section, University of Bari “Aldo Moro”, 70121 Bari, Italy

**Keywords:** EMT, Sjögren’s syndrome, Follistatin-like 1 protein, TGF-β1, SMAD2/3

## Abstract

Background: The activation of the epithelial to mesenchymal transition (EMT) program is a pathological response of the Sjögren’s syndrome (SS) salivary glands epithelial cells (SGEC) to chronic inflammation. Follistatin-like 1 protein (FSTL1) is a secreted glycoprotein induced by transforming growth factor-β1 (TGF-β1), actively involved in the modulation of EMT. However, the role of FSTL1 in the EMT program activation in SS has not yet been investigated. Methods: TGF-β1-stimulated healthy human SGEC, SS SGEC, and SS salivary glands (SGs) biopsies were used to assess the effect of FSTL1 on the activation of the EMT program. FSTL1 gene activity was inhibited by the siRNA gene knockdown technique. Results: Here we reported that FSTL1 is up-regulated in SS SGs tissue in a correlated manner with the inflammatory grade. Blockage of FSTL1 gene expression by siRNA negatively modulates the TGF-β1-induced EMT program in vitro. We discovered that these actions were mediated through the modulation of the SMAD2/3-dependent EMT signaling pathway. Conclusions: Our data suggest that the TGF-β1-FSTL1-SMAD2/3 regulatory circuit plays a key role in the regulation of EMT in SS and targeting FSTL1 may be a strategy for the treatment of SGs EMT-dependent fibrosis.

## 1. Introduction

Follistatin-like protein 1 (FSTL1) is a secreted glycoprotein with extensive glycosylation modifications, produced primarily by cells of the mesenchymal phenotype [1,2,3]. FSTL1 is composed of the presence of a follistatin (FS)-like domain and an extracellular calcium-binding (EC) motif and belongs to the SPARC (secreted protein acidic and rich in cysteine) family of matricellular proteins whose members participate in fine cellular functions [4]. Although the role in the progression of most pathological processes remains unclear, recent studies have described the involvement of FSTL1 in several pathological and physiological processes, such as embryonic development, tissue remodeling/repair, and organ fibrosis [5,6].

Converging lines of evidence have suggested an intrigant key role of FSTL1 in epithelial-mesenchymal transition (EMT) [7], a dynamic and reversible process in which cells gradually lose their epithelial phenotype and transform into a mesenchymal phenotype; moreover, sustained activation of EMT, in the context of the response to injury, promotes inflammation, triggering the fibrotic pathology of multiple organs [8,9]. Interesting studies have underlined as FSTL1 promotes EMT in concert with TGF-β1 [10,11]. Indeed, TGF-β1 regulates epithelial injury, myofibroblasts proliferation and differentiation, collagen production, and deposition. For this reason, it is indicated as a “master switch” in the initiation of the cascade of events that characterizes the EMT-dependent organ fibrosis [12,13,14,15].

Interestingly, there is additional evidence describing FSTL1 as a downstream effector of TGF-β1-induced fibrotic responses, and it is demonstrated that FSTL1 is upregulated in several fibrotic tissues, and promotes fibrogenesis through facilitating TGF-β1 signaling [10,16].

In recent years, a number of published reports have identified a new role for FSTL1 in the regulation of immune cell function, demonstrating overexpression of FSTL1 in several autoimmune diseases [16,17]. An interesting study performed by Li and colleagues [16] demonstrated that by targeting FSTL1, the attenuation of bleomycin-induced pulmonary and dermal fibrosis in vivo and TGF-β1-induced dermal fibrosis ex vivo in human skin was detected, a finding also confirmed by experimental models. In addition, elevated serum levels of FSTL1 were detected in patients with rheumatoid arthritis, and Sjögren’s syndrome (SS), levels which are much higher than the levels observed in other chronic inflammatory diseases such as ulcerative colitis, systemic lupus erythematosus, systemic sclerosis, and polymyositis/dermatomyositis which, however, show altered levels of FSTL1 [17]. According to these literature data, elevated FSTL1 serum levels were detected in patients affected by primary Sjögren’s syndrome (pSS) [16,17]. pSS is a chronic, systemic autoimmune disorder, commonly linked with dry eyes and dry mouth [18] and characterized by the trigger of the TGF-β1/SMAD-dependent EMT process in response to chronic inflammatory factors release [13].

In fact, the role of FSTL1 in the pathogenic pSS EMT has not yet been investigated [10,11,16]. In this report, we analyze the expression of FSTL1 in pSS salivary glands (SGs) biopsies and study the induction of FSTL1 by TGF-β1 treatment in human healthy salivary gland epithelial cells (SGEC); in addition, we evaluate the ability of FSTL1 to activate SMAD2/3-regulated TGF-β1-dependent EMT cascade in pSS.

## 2. Materials and Methods

### 2.1. Cell Treatment and Antibodies

Recombinant human TGF-β1 was purchased from R&D Systems (Minneapolis, MN, USA). Mouse anti-human TGF-β1 monoclonal antibody (mAb) (1:50, Santa Cruz Biotechnology, Santa Cruz, CA, USA), goat anti-human SMAD2/3 polyclonal Ab (pAb) (1:100, R&D Systems, Minneapolis, MN, USA), goat anti-humanp-SMAD2/3 (Ser 423/425) pAb (1:100, Santa Cruz Biotechnology), mouse anti-human β-actin mAb clone AC-15 (1:100, Sigma-Aldrich, St. Louis, MO, USA), mouse anti-human E-cadherin mAb (1:100, Dako, Santa Clara, CA, USA), mouse anti-human vimentin mAb (1:100, Thermo Fisher Scientific, Waltham, MA, USA), human Follistatin-like 1/FSTL1 polyclonal Antibody (pAb) (1 μg/mL, R&D Systems) were used in the experimental procedure.

### 2.2. Bioptic Samples Collection

The Department of Pathology, University of Bari Medical School, selected 30 labial SGs biopsies from pSS patients (aged 66.8 ± 2.3 years) according to the ACR/EULAR classification criteria for pSS [19], (using an absolutely anonymous form). The slides relating to 20 biopsies were already present in the archive. In addition, slides, and small glandular fragments from 10 SS patients who had to undergo a biopsy for diagnostic purposes were collected, maintaining anonymity. Five items were evaluated for the selection: anti-SSA/Ro antibody positivity and focal lymphocytic sialadenitis with a focus score of ≥1 foci/4 mm^2^, each scoring 3, and an abnormal ocular staining score of ≥5 (or van Bijsterveld score of ≥4), a Schirmer’s test result of ≥5 mm/5 min, and an unstimulated salivary flow rate of ≥0.1 mL/min, each scoring 1. On the basis of these parameters, biopsies were subdivided into three inflammatory groups: I, low; II, intermediate; III, advanced. Healthy volunteers (aged 60.1 ± 1.4 years, *n* = 10) with no salivary condition were included in the study as controls. Controls are healthy individuals awaiting the removal of salivary mucoceles from the lower lip [20]. The healthy subjects had no complaints of oral dryness, no autoimmune disease, and normal salivary function. LSG samples were collected from the lower lip under local anesthesia through normal mucosa. The cell cultures of SS SGEC were obtained using pieces of biopsy samples from SGs biopsies, a process which the patients underwent for diagnostic purposes, and the patients agreed that we could use a small piece of the gland to prepare the cultures. Similarly, the cells of healthy subjects were obtained from individuals who underwent the removal of the glands for mucoceles and not for purely experimental purposes.

### 2.3. Histochemistry

Representative serial 3 μm sections of healthy and pSS formalin-fixed, paraffin-embedded minor SG (MSG) tissues were rehydrated and deparaffinized in graded alcohol. (1 h in 70% ethanol supplemented with 0.25% NH_3_ and immersion in 50% ethanol for 10 min). The slides were washed in phosphate-buffered saline (PBS) (pH 7.6 3 × 10 min) and immersed in EDTA buffer (0.01 M, pH 8.0) for 20 min to unmask antigens. The immunolabeling was performed by blocking the endogenous peroxidase by treatment with 3% hydrogen peroxide solution in water for 10 min at room temperature (RT) and carrying out a preincubation in non-immune donkey serum (Dako LSAB Kit, Dako, CA, USA). Then, the slides were incubated overnight at 4 °C with primary Abs. The relative secondary Abs (Santa Cruz Biotechnology, Dallas, TX, USA) diluted 1:200 in PBS for 1 h at RT followed by the streptavidin-peroxidase complex (Vector Laboratories, Newark, CA, USA) for 1 h at RT were applied to the sections. Then, 3,3-diaminobenzidine tetrahydrochloride (DAB) was used as chromogen (Vector Laboratories), and hematoxylin (Merck Eurolab, Dietikon, Switzerland) for counterstaining. Negative controls of the immunoreactions were performed by replacing the primary Ab with donkey serum. After the addition of the secondary Ab, no specific immunostaining was observed in the negative controls.

### 2.4. FSTL1 Immunohistochemical Analysis and Quantification

For immunohistochemistry image quantification, representative areas of the sections were viewed using a ×20 objective and photographed using a Zeiss Axio Cam HRc (Zeiss, Oberkochen, Germany). For each image of stained slides, ten representative visual fields, each 586 μm × 439 μm in area, were randomly acquired by the computerized image-acquisition (ImageJ, version 1.46c; WS Rasband, National Institutes of Health, Bethesda, MD, USA) connected to the microscope. SGEC were identified on the captured images and the number of SGEC positive for FSTL1 and the area occupied by these cells were measured. Positive areas were expressed as a percentage of the total tissue area examined.

### 2.5. SGEC Culture and Treatment

Healthy and pSS glandular fragments belonging to each classification group were dissociated by enzymatic and mechanical means into a suspension of single cells and plated onto a culture flask. Cells were used to obtain the experimental material (RNA and proteins) used to carry out our evaluations. After dissociation pSS dispersal cells were resuspended in McCoy’s 5a modified medium supplemented with 10% heat-inactivated (56 °C for 30 min) FCS, 1% antibiotic solution, 2 mM L-glutamine, 50 ng mL^−1^ epidermal growth factor (EGF, Promega, Madison, WI, USA), and 0.5 μg mL^−1^ insulin (Novo, Bagsværd, Denmark) and incubated at 37 °C, 5% CO_2_ in the air. The contaminating fibroblasts were selectively removed using 0.02% EDTA treatment. Immunocytochemical confirmation of the epithelial origin of cultured cells was routinely performed, as previously described [21]. Healthy human SGEC were grown in the same modified McCoy’s 5A medium (Invitrogen, Waltham, MA, USA) supplemented with 1% heat-inactivated FCS (to avoid excessive growth during treatment with TGF-β1). Healthy SGEC were stimulated with 10 ng/mL of TGF-β1 in the growth medium for 24–48 h and then harvested for FSTL1 analysis. To inhibit FSTL1, siRNA gene knockdown technique was used. All experiments were performed in triplicate and repeated three times.

### 2.6. Reverse Transcriptase Polymerase Chain Reaction (RT-PCR) and Quantitative Real-Time PCR (q-RT-PCR)

Total RNA from all experimental samples was isolated using the TRIzol reagent (Invitrogen Corp., Waltham, MA, USA). First-strand cDNA was synthesized by M-MLV reverse transcriptase (Promega, Madison, WI, USA) with 1 μg each of DNA-free total RNA sample and oligo-(dT)15 (Life Technologies, Grand Island, NY, USA). Equal amounts of cDNA were subsequently amplified by PCR in a 20 μL reaction mixture containing 2 μM of each sense and antisense primer, PCR buffer, 2.4 mM MgCl2, 0.2 mM each dNTP, 10 μL of transcribed cDNA, and 0.04 U/μL Taq DNA polymerase. The primers used to amplify cDNA fragments were as follows: 5′-GGGAACTGCTGGCTCC-3′ and 5′-TTTACAGGGGATGCAG-3′ were used for FSTL1 gene amplification, SMAD2, forward 5′-ACTAACTTCCCAGCAGGAAT-3′ and reverse 5′-GTTGGTCACTTGTTTCTCCA-3′; SMAD3, forward 5′-ACCAAGTGCATTACCATCC-3′ and reverse 5′-CAGTAGATAACGTGAGGGAGCCC-3′; E-cadherin, forward 5′-TTCCCTGCGTATACCCTGGT-3′ and reverse 5′-GCGAAGATACCGGGGGACACTCATGAG-3′; vimentin, forward 5′-AGGAAATGGCTCGTCACCTTCGTGAATA-3′ and reverse 5′-GGAGTGTCGGTTGTTAAGAACTAGAGCT-3′. The PCR cycling profile consisted of an initial denaturation step at 95 °C for 15 min, followed by 35 cycles of 94 °C for 60 s, 58 °C for 60 s, and 72 °C for 60 s. Amplification products were run on 1.5% agarose gel soaked in ethidium bromide and visualized under ultraviolet transillumination. The reference gene for the analysis was Glyceraldehyde 3-phosphate dehydrogenase (GAPDH). For qRT-PCR, forward and reverse primers for all the genes tested and the internal control gene β-2microglobulin (part n° 4326319E; β2M) were purchased from Applied Biosystems (Assays-On-Demand, Applied Biosystems, Waltham, MA, USA). The instrument used for amplification was ABI PRISM 7700 sequence detector (Applied Biosystems). The 2^−ΔΔCT^ method was used for calculating relative gene expression values in qPCR.

### 2.7. Data Evaluation and Sequence Analysis

mRNA expression was quantified as the average of a set of three independent experiments, performed by gel image software (Bio-Profil Bio-1D; LTF Labortechnik GmbH, Wasserburg, Germany), and the GAPDH-related intensity for each band was expressed as arbitrary units. The identity of each PCR product was confirmed by the size, and the direct sequencing using the gene-specific forward or reverse primers.

### 2.8. Western Blot Analysis

Treated and untreated control SGEC were placed in a homogenization buffer (200 μL) containing 1% Triton X-100, 50 mM Tris-HCl (pH 7.4), 1 mM PMSF, 10 μg/mL soybean trypsin inhibitor, and 1 mg/mL leupeptin. After centrifugation, Bradford’s protein assay was performed to determine protein concentrations. Electrophoresis on 10% SDS-polyacrylamide gels was performed, followed by blotting at 200 mA (constant amperage) and 200 V for 110 min. Blots were blocked, washed three times with 0.1% (*v*/*v*) Tween 20-PBS 1× (T-PBS) and membranes were probed with the appropriate primary antibodies listed above. Chemiluminescence was revealed according to the protocol (Santa Cruz Biotechnology). Mouse anti-human β-actin mAb clone AC-15 (1:100, Sigma-Aldrich, St. Louis, MO, USA; 0.25 μg/mL) was used as a protein loading control.

### 2.9. FSTL1 siRNA Transfection

The siRNA sequences used for down-regulation of human FSTL1 were designed and synthesized by Thermo Fisher Scientific. An irrelevant siRNA with random nucleotides and no known specificity was used to normalize relative gene inhibition of the target gene. Cells were transfected with siRNA for FSTL1 using the siPORT NeoFX transfection agent (Ambion, Austin, TX, USA) accordingly to the manufacturer’s manual. A mixture of siRNA duplexes was used to obtain the highest efficiency. GAPDH siRNA primers were used as positive controls (Ambion), and a negative control with no known sequence similarity to human genes was included. Silencing was observed by mRNA quantification by qRT–PCR.

### 2.10. Statistical Analysis

Statistical analysis was performed by calculation of the mean percentage ± SE of data obtained from a minimum of three experiments. Differences among groups were determined using *t*-test. Statistical significance was set at * *p* < 0.05 and ** *p* < 0.01.

## 3. Results

### 3.1. Aberrant Expression of FSTL1 in SS SGs Is Associated with Inflammatory Grade 

We examined the expression of FSTL1 by immunohistochemistry (IHC) in pSS biopsy samples, in comparison with healthy subjects. The number and the distribution of lymphocytic foci in the different pSS SGs bioptic specimens were quantified and then classified as I, low; II, intermediate; and III, advanced, respectively. Ten biopsy specimens for each group were analyzed. Anti-human FSTL1 antibody pAb was used to evidence SGs FSTL1 expression (Figure 1). As shown in Figure 1, the staining of FSTL1 protein ranged from weak to strong, and the results showed that FSTL1 positive staining was increased in those biopsies characterized by a higher inflammatory degree. As observed FSTL1 acinar cell expression was weak and, in all glandular specimens, FSTL1 seems to be expressed more in the epithelial cells of the ducts than in those of the acini (panels a and b. In addition, the experimental procedures clearly showed a different intensity of expression between glands of healthy and diseased subjects: the intensity of FSTL1 expression detected at the level of the ducts is much more pronounced in the biopsy of pSS subjects than in healthy control glands (Figure 1a,b). Absorbance measurements performed by ImageJ software (ImageJ, version 1.46c; WS Rasband, National Institutes of Health, Bethesda, MD, USA) (Figure 1b) confirmed this observation and highlights how the intensity of expression increases with the inflammatory degree showing that staining for FSTL1 was significantly darker in pSS glands, grade III, than in the control glands (93.9 ± 2.34 vs. 15.8 ± 1.43, *p* < 0.01). 

### 3.2. TGF-β1 Upregulates FSTL1 Expression in Healthy Salivary Glands Epithelial Cells

Healthy SGEC were treated with 10 ng/mL of TGF-β1 for 24 and 48 h. Figure 2A,B, shows that the TGF-β1 treatment resulted in the upregulation of FSTL1 mRNA expression (detected after a 24-h treatment) (*p* < 0.01) in comparison with untreated cells. These results were confirmed by Real-Time PCR as reported in Figure 2C. Regarding the FSTL1 protein expression, it significantly increased in the healthy SGEC were significantly increased after a 48-h treatment with TGF-β1 (*p* < 0.01) (Figure 2D,E). As a positive control, FSTL1 was also analyzed directly in pSS SGEC, confirming an overexpression of the FSTL1 gene and protein (Figure 2A–E).

### 3.3. FSTL1 Neutralization Attenuates EMT-Related Morphological Changes Induces by TGF-β1 Treatment

As previously demonstrated and microscopically revealed, TGF-β1 has been implicated as a primary inducer of EMT in SS [13]. To evaluate the influence of overexpression of FSTL1 on TGF-β1-induced EMT and examine the impact of FSTL1 expression on TGF-β1-induced morphological changes in SGEC, we decide to transiently silence the expression of the FSTL1 gene with siRNA in healthy SGEC before stimulation with TGF-β1. SGEC cell morphology was microscopically evaluated. Following treatment with TGF-β1 for a maximum of 72 h, primary healthy SGEC exhibited a marked alteration in cell morphology, changing from the characteristic organized ‘cobblestone’ appearance of differentiated epithelial cell monolayers to a disorganized elongated fibroblast-like phenotype (Figure 3c). In particular, TGF-β1-treated SGEC lost their junctions and, consequently, their polarized epithelial phenotype and acquired elongated mesenchymal traits, becoming dispersed and showing a fibroblast-like morphology with a front/back polarity. FSTL1 siRNA transfection significantly inhibited TGF-β1-induced morphologic changes in SGEC. After FSTL1 gene silencing, these cells maintained the morphologic characteristic of epithelial cells and showed no evidence of a transformation towards mesenchymal phenotype (Figure 3d,e). Based on these phenomena, we speculated that FSTL1 is induced in response to TGF-β1 treatment and is causally involved in driving the EMT program.

### 3.4. FSTL1 Moderates EMT-Related Mesenchymal Markers Expression in TGF-β1-Treated SGEC

We next investigated whether diminished epithelial cell activation in TGF-β1-treated healthy SGEC after the FSTL1 gene knockdown, altered the expression of mesenchymal and epithelial markers, often used in the evaluation of EMT activation. We reasoned that TGF-β1-treated glandular epithelial cells may underwent an EMT program activation through FSTL1 overexpression which may influence epithelial and mesenchymal factors modulation. Healthy SGEC were treated with 10 ng/mL of TGF-β1 after the transfection with FSTL1 siRNA, and the expression of the gene biomarkers of EMT was examined by RT-PCR and quantitative Real-time PCR. As shown in Figure 4A–C, TGF-β1-treated SGEC shows reduced E-cadherin gene expression accompanied by upregulation of gene mesenchymal marker vimentin, but the FSTL1 gene expression inhibition significantly reverses the situation by blocking mesenchymal gene transcription (*p* < 0.01) (Figure 4A–C). We also detected these markers’ expression at protein’s levels. By Western blot analysis, we found that the level of EMT protein markers significantly changed following 10 ng/mL of TGF-β1 in cells transfected with FSTL1 siRNA for 48 h (Figure 5A,B). In fact, blocking the effect of TGF-β1-dependent release of FSTL1 by the gene silencing technique, the expression of E-cadherin was increased, and vimentin was decreased in a significant manner (*p* < 0.01) (Figure 5A,B). As control in these experiments, we also used SS SGEC treated with FSTL1 siRNA in vitro, which confirm genes and proteins results obtained in TGF-β1-trated healthy SGEC, demonstrating that FSTL1 could promote EMT in SS SGEC.

### 3.5. FSTL1 Modulates EMT by Facilitating TGF-β1 Signaling through SMAD2/3 Phosphorylation and Activation

To evaluate the roles of FSTL1 in the TGF-β1 signaling cascade, we blocked FSTL1gene expression by FSTL1 siRNA in human SGEC treated with TGF-β1. As already demonstrated, activation of the TGF-β1-SMAD2/3 pathway by TGF-β1 treatment resulted in phosphorylation and activation of SMAD2 and SMAD3 [13]. Interestingly, we observed that FSTL1 gene silencing severely impaired (*p* < 0.01) SMAD2 and SMAD3 phosphorylation in response to TGF-β1 stimulation (Figure 5A,B). The inhibition of phosphorylation of SMAD2/3 proteins, observed after FSTL1 gene expression downregulation, was confirmed also for SMAD2 and SMAD3 gene analysis performed by RT-PCR and Real-time PCR (Figure 4A–C). These results indicate that FSTL1 is a crucial component of TGF-β1 signaling in SGEC contexts.

## 4. Discussion

SS is a lifelong chronic inflammatory autoimmune disorder affecting primarily the lachrymal and SGs [22]. In this disease, the glandular secretory activity is reduced resulting in dryness in the eyes, mouth, and throat. In addition, often, SS dysfunction involves other organs, along with complications such as pain, fatigue, and digestive problems [22]. A targeted and early diagnosis is necessary to develop more effective therapies. The major obstacle to overcome in order to identify therapeutics is a lack of complete understanding of the molecular mechanisms underlying the pathogenesis of this disease; SS seems, in fact, multifactorial, but a key role is certainly played by the chronic release of inflammatory mediators. The findings reported here add new insights into the knowledge of SS pathogenesis and consolidate the hypothesis of the role of FSTL1 in promoting EMT. 

EMT is a process of dedifferentiation and transformation of epithelial cells into mesenchymal cells [23]. During EMT, the expression of epithelial markers such as E-cadherin is decreased, while the expression of mesenchymal markers, for example, vimentin, is up-regulated [24]. The EMT program foresees the triggering of a cascade of events in which various factors are activated, and among these, the phosphorylation of SMADs represents a crucial event [25,26]. Cells that undergo EMT show a phenotypic switch from epithelial cells to fibroblastic or mesenchymal cells [23].

EMT may drive inflammatory reactions and, in turn, is influenced by the inflammatory microenvironment [27,28,29]. Based on these characteristics, EMT is a key process in determining the onset of fibrotic phenomena that lead to the loss, or, at least, to the alteration of the functionality of the affected organs [30,31,32]. Therefore, identifying molecules that can inhibit EMT could help reduce the tendency to evolve toward fibrosis. 

In this scenario, the Follistatin Like (FSTL) family of proteins fits well. FSTL proteins are involved in cell migration, proliferation, and cellular differentiation, through the binding to the activin protein of the TGF-β family, which represents key activators of the EMT program [6]. Recently, pioneering studies conducted on human SGEC have led to very intriguing results; they demonstrated that, in SS, TGF-β activates EMT by both the canonical SMAD2/3 pathway and the non-canonical MAPK pathway influenced by the release of pro-inflammatory cytokines in a condition of chronic inflammation. The EMT program includes the activation of Snail, a transcription factor, and determines, consequently, an imbalance in favor of the expression of mesenchymal markers compared to epithelial markers. Interestingly, the activation of the EMT cascade leads to a dramatic evolution toward fibrosis [13,29,32].

In the experimental study reported here, we wanted to analyze the relationship between FSTL1 and EMT in SS, since elevated serum levels of FSTL1 were already detected in the serum of SS patients [17]. For the first time, we detected the expression of FSTL1 in pSS SGs biopsies. IHC analysis showed that the expression of FSTL1 was positively correlated with the inflammatory grade, and, based on the correlation between inflammation and EMT, this led us to evaluate, as a second step, the correlation between FSTL1 and EMT.

Bright field microscopic evaluation demonstrated that by blocking the activity of FSTL1 by its gene knockdown, human healthy SGEC treated with TGF-β1 as an EMT inducer, did not show any morphological changes: on the contrary, cells treated with TGF-β1 alone showed mesenchymal phenotypes, as already demonstrated [13]. Moreover, using healthy SGEC in vitro-treated with TGF-β1 and transfected with FSTL1 siRNA, we demonstrated that the over-expression of FSTL1 down-regulated the expression of E-cadherin and up-regulated the expression of vimentin, confirming that FSTL1 could promote EMT. To further explore our findings of a role for SMAD2/3 activation in the FSTL1-induced EMT process in SS, we examined healthy SGEC treated with TGF-β1 following or not FSTL1 siRNA transfection, demonstrating that cells in which FSTL1 is active show an extremely high expression of p-SMAD2/3 similar to the levels of SMAD2/3 phosphorylation observed in SS SGEC, utilized as a useful positive control. Results obtained identified FSTL1 as a component of the TGF-β-SMAD2/3 pathway that stimulates SMAD2/3 phospho-activation in SS. In conclusion, our observations suggest that interactions between FSTL1 and TGF-β1 signal transduction pathways may determine a cascade of mechanisms activation by which alterations in p-SMAD2/3 expression occur, and neutralization of FSTL1 gene activity had inhibitory effects on the phosphorylation of SMAD2/3 induced by TGF-β1; this effect, consequently, attenuates EMT (Figure 6). Our findings suggest a promising therapeutic approach for SGEC EMT in SS. However, all experimental research in the present study was performed in vitro, and, actually, there were no in vivo data supporting these results. Hence, further research in SS patients or animal models needs to be performed in the future to confirm the therapeutic potential of FSTL1.

## Figures and Tables

**Figure 1 jcm-11-05368-f001:**
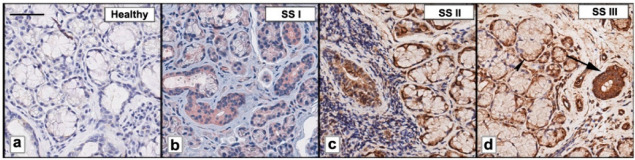
FSTL1 was immunohistochemically evaluated in pSS biopsy samples classified as I, low; II, intermediate; and III (**b**–**d**). Healthy SGs bioptic specimen was used as control (**a**). As observed, FSTL1 acinar cell expression was weak while FSTL1 staining was more pronounced in ductal epithelium (**b**–**d**); furthermore, a positive correlation was found between FSTL1 staining and inflammatory degree (**b**–**d**). Brown staining shows positive immunoreaction; blue staining shows nuclei. Bar = 20 μm. Arrows show FSTL1 distribution in ducts (large arrows) and acini (small arrows).

**Figure 2 jcm-11-05368-f002:**
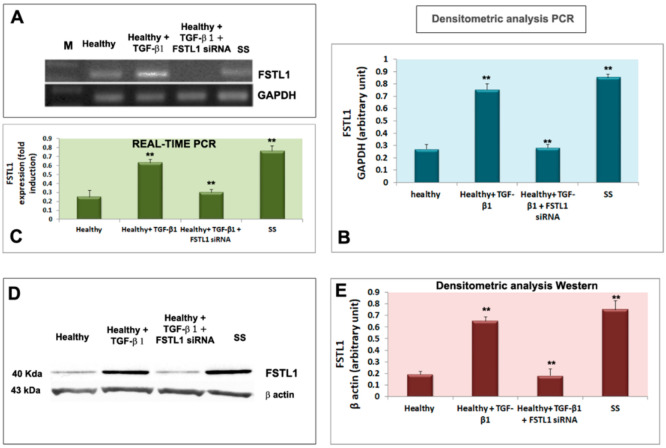
TGF-β1 treatment resulted in the upregulation of FSTL1 expression in healthy SGEC treated with TGF-β1. (**A**,**B**) show that the TGF-β1 treatment resulted in the upregulation of FSTL1 mRNA expression compared to untreated cells, with GAPDH as a reference gene. Quantitative Real-Time PCR confirmed this result (**C**). Normalized gene expression levels were given as the ratio between the mean value for the target gene and that for the β-2 microglobulin. PCR reactions were performed in triplicate and the data were presented as fold changes in gene expression (mean ± SE of three independent experiments). FSTL1 protein expression was significantly induced by TGF-β1 treatment as demonstrated by immunoblotting (**D**,**E**); (** = *p* < 0.01).

**Figure 3 jcm-11-05368-f003:**
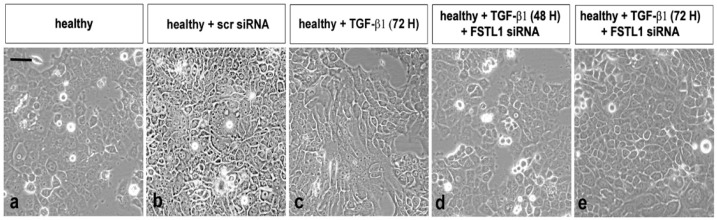
Bright-field micrographs of SGEC treated with TGF-β1 after FSTL1 gene silencing. SGEC cell morphology was microscopically evaluated. FSTL1 gene knockdown significantly inhibited TGF-β1-induced morphologic changes in SGEC (**e**). After transfection with FSTL1 siRNA, SGEC show a more epithelial-like morphology in comparison with SGEC treated with TGF-β1 alone for 48–72 h (**d**,**e**). (**a**): untreated control cells; (**b**): healthy SGEC transfected with scrambled control siRNA; (**c**): TGF-β1 treated SGEC for 72 h.

**Figure 4 jcm-11-05368-f004:**
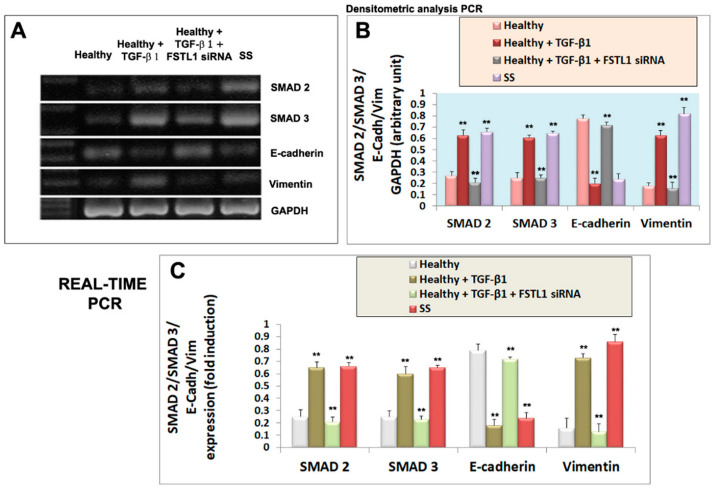
Effect of FSTL1 gene silencing on E-cadherin, vimentin, SMAD2 and SMAD3 gene expression evaluated by RT-PCR (**A**,**B**) and quantitative Real-Time PRC (**C**). (**A**): E-cadherin, vimentin, SMAD2, and SMAD3 gene expression were determined by RT-PCR in healthy SGEC (control, lane 1) cultured for 24 h with 10 ng/mL of TGF-β1 in which FSTL1 gene silencing was induced (lanes 2, 3). SS SGEC mRNA was used as a positive control (lane 4). (**B**) represent the densitometric analysis. Real-time PCR conducted on SGEC submitted to some treatments was shown in (**C**). GAPDH (**A**,**B**) and β-2-microglobulin (**C**) were used as reference genes for mRNA analyses. Values represent the mean ± SE of three independent experiments, with ** = *p* < 0.01.

**Figure 5 jcm-11-05368-f005:**
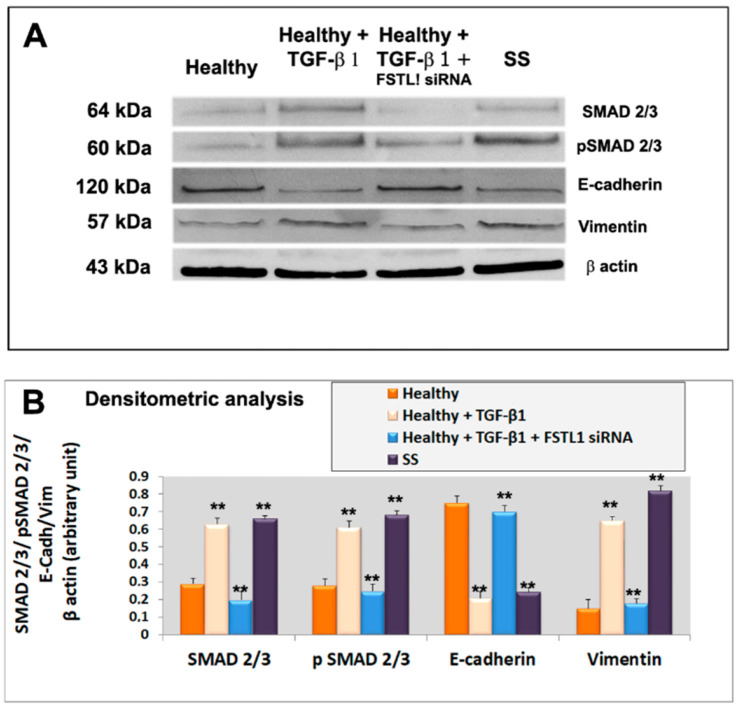
(**A**): representative Western blot analysis of E-cadherin, vimentin, total SMAD2/3 and P-SMAD2/3 proteins expression in healthy untreated control SGEC (lane 1), healthy SGEC treated with TGF-β1 for 48 h (lane 2), cells transfected with FSTL1 siRNA and subsequently treated with TGF-β1 for 48 h (lane 3). SS SGEC protein lysate was used as a positive control. For quantitative analysis banding densities were compared with test marker β-actin (**B**). Data are expressed as a significant change relative to the untreated control cells. Each bar represents the mean ± SE. **, *p* < 0.01. Each experiment was repeated three times.

**Figure 6 jcm-11-05368-f006:**
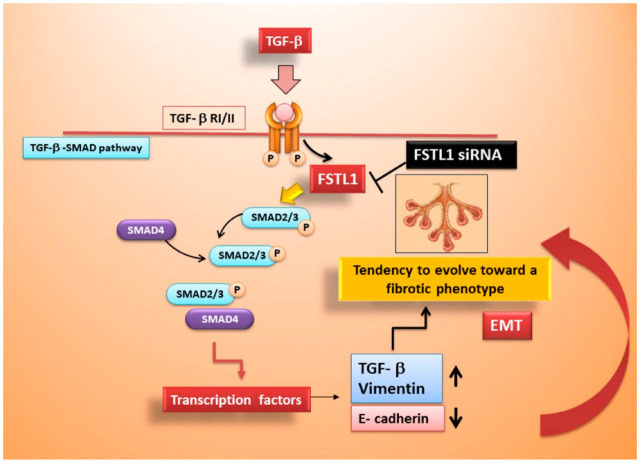
Schematic representation of FSTL1 role in TGF-β1-induced EMT in SS.

## Data Availability

All relevant data are within the paper.

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
