# Peer review of "The Expression of Follistatin-like 1 Protein Is Associated with the Activation of the EMT Program in Sjögren’s Syndrome"

_jcm, 2022, doi:10.3390/jcm11185368_

Round 1

Reviewer 1 Report

This study reports the suspected role of Follistatin-like-1 protein (FSTL1) in the activation of the epithelial to mesenchymal transition (EMT) program. The EMT program is involved in the pathogenesis of Sjögren’s syndrome (SS).

The study was well conducted. The results concerning FSTL1 upregulation and EMT program modulation, including the involvement of SMAD2/3 pathways, seem strong to me, since there are supported by different approaches reporting the same conclusions (mRNA expression, qRT-PCR, immunoblot or westernblot). All together, these results bring strong evidence on the suspected role of FSTL1 in the immune and morphological changes observed in salivary glands of pSS patients, at least in vitro.

Major comments:

-       The introduction is clear and summarizes what is known about FSTL1, its relation with inflammatory process and its relation with TGF-B1-induced fibrosis. However, it does not seem appropriate to me that the authors give the summarized results of their study at the end of the introduction (line 59/60).

-       Methods (line 80): I did not understand how were classified the patients between the 3 groups, according to the scores given line 77 and line 79 (thresholds?). According to line 221 (legend of figure 1), it sounds that the classification was only performed on the results of focus scores?

-       Line 85: who were the healthy patients? Did they have a sicca syndrome? Were there patients with cancer? How many healthy patients/biopsies were included (line 199). Those details should appear in the Methods section or in the Results.

-       Ethical issues: the IRB number for this study should be provided.

-       Figure 2: the number of patients in each subgroup (healthy, healthy+TGF-B, etc…) should be detailed on the figure or in the legend.

-       Figure 4: the number of patients in each subgroup (healthy, healthy+TGF-B, etc…) should be detailed on the figure or in the legend.

-       Figure 6: the figure is didactic and summarizes most of the result. However, I would suggest to report that the results came from in vitro studies. Moreover, I’d suggest to replace “salivary gland fibrosis” by “tendency to evolve toward a fibrotic phenotype” (or something more moderate).

Minor comments:

-       Introduction (line 51): Maybe the authors could give more details on the type of autoimmune diseases where the role of FSTL1 have been reported (ref 16 and 17).

-       Line 74: separate pSS and labial

-       Line 74: please reword, so that we can understand that these 30 biopsies come from 30 patients

-       Line 216 to 218: comments not appropriate in the manuscript?

-       Line 233/234: typo (rephrase)

-       Line 322 to 325: These first lines of the discussion do not seem to be correlated to the topic of the article (mechanistic pathways of the disease rather than clinical symptoms/quality of life). I’d suggest to rephrase that.

-       Discussion is interesting and based on the results from the study. However, a double check of English grammar and syntax would help to improve the quality of this part.

Author Response

We would like to express our sincere gratitude to the reviewer for his/her constructive and positive comments and for the very thoughtful critique of our manuscript and we are pleased to say that we tried to address all the concerns raised. All changes to the manuscript are highlighted in the text. We respond below in detail to each of the reviewer’s comments and we hope that the reviewer will find satisfactory our responses to his comments. 

Major comments:

-       The introduction is clear and summarizes what is known about FSTL1, its relation with inflammatory process and its relation with TGF-B1-induced fibrosis. However, it does not seem appropriate to me that the authors give the summarized results of their study at the end of the introduction (line 59/60).

The final sentence has been modified in accordance with the suggestion

-       Methods (line 80): I did not understand how were classified the patients between the 3 groups, according to the scores given line 77 and line 79 (thresholds?). According to line 221 (legend of figure 1), it sounds that the classification was only performed on the results of focus scores?

I would like to specify that the biopsies derive from patients classified using the 5 items reported in the materials and methods. According to your observation, the phrase in the legend may be misleading and has been deleted.

-       Line 85: who were the healthy patients? Did they have a sicca syndrome? Were there patients with cancer? How many healthy patients/biopsies were included (line 199). Those details should appear in the Methods section or in the Results.

As already reported in the materials and methods, paragraph 2.2 "Patient Selection and Characteristics", now renamed “2.2. Bioptic samples collection”, the healthy volunteers considered for the study were 10. We added some information to clarify the healthy volunteers’ characteristics;  we reported that: “They are healthy individuals awaiting removal of salivary mucoceles from the lower lip. The healthy subjects had no complaints of oral dryness, no autoimmune disease and had normal salivary function. LSG samples were collected from the lower lip under local anesthesia through normal mucosa”.

-       Ethical issues: the IRB number for this study should be provided.

Thank you for your observation: I would to precise that our experimental procedure does not include any interaction or intervention with human subjects (exclusively finalized to research) or include any access to identifiable private information, then IRB approval is not relevant to this content.

-       Figure 2: the number of patients in each subgroup (healthy, healthy+TGF-B, etc…) should be detailed on the figure or in the legend.

-       Figure 4: the number of patients in each subgroup (healthy, healthy+TGF-B, etc…) should be detailed on the figure or in the legend.

In accordance with your objections related to figures 2, 4, I try to clarify the experimental procedures that have allowed us to carry out immunohistochemistry and in vitro studies on cell cultures. The Department of Pathology, selected, in total, 30 labial SGs biopsies from pSS patients (in an absolutely anonymous form). The slides relating to 20 biopsies were already present in the archive. In addition, 10 SS patients had to undergo a biopsy for diagnostic purposes and we agreed with the Department of Pathology to receive some slides and a small fragment of gland to perform our experiments on cell cultures, after received their oral consent. I repeat that the biopsy was not carried out for experimental purposes, but for diagnostic purposes independent of our experimental activity and we received the material in an absolutely anonymous form (this is why IRB approval is not relevant to this content). All the bioptic slides were classified into the 3 groups described in Materials and Methods and, to set up cell cultures, the glandular fragments obtained, similarly, were divided into the 3 groups described as I, [low (N = 4); II, intermediate (N = 3); and III, advanced (N = 3)]. The glandular fragments belonging to each group were dissociated by enzymatic and mechanical means into a suspension of single cells and plated onto a culture flask. Cells were used to obtain the experimental material (RNA and proteins) used to carry out our evaluations.

-       Figure 6: the figure is didactic and summarizes most of the result. However, I would suggest to report that the results came from in vitro studies. Moreover, I’d suggest to replace “salivary gland fibrosis” by “tendency to evolve toward a fibrotic phenotype” (or something more moderate).

We agree with your comment and have modified the figure 6 in accordance with your suggestion.

Minor comments:

-   Introduction (line 51): Maybe the authors could give more details on the type of autoimmune diseases where the role of FSTL1 have been reported (ref 16 and 17).

As suggested, we have added some data related to references 16, 17

-       Line 74: separate pSS and labial

Done

-       Line 74: please reword, so that we can understand that these 30 biopsies come from 30 patients Done

-       Line 216 to 218: comments not appropriate in the manuscript?

We agree with your comment and have deleted the sentence.

-       Line 233/234: typo (rephrase)

We rewrote the sentence.

-       Line 322 to 325: These first lines of the discussion do not seem to be correlated to the topic of the article (mechanistic pathways of the disease rather than clinical symptoms/quality of life). I’d suggest to rephrase that.

As suggested we rephrase the sentence.

-       Discussion is interesting and based on the results from the study. However, a double check of English grammar and syntax would help to improve the quality of this part.

We double-checked the discussion for grammar and syntax hoping to have fixed all errors and improved it.

Reviewer 2 Report

This study discussed the role of FSTL1 in the pathogenic pSS EMT. It is a valuable research that FSTL1 triggers SMAD2/3 phosphorylation in the activation of TGF-β1-dependent EMT cascade in pSS. However, I have some concerns about this study.

1.     Follistatin-like protein 1 (FSTL1) is a secreted glycoprotein with extensive glycosylation modifications, produced primarily by cells of the mesenchymal phenotype. In this study, the data showed that FSTL1 expression in acinar cells and ductal epithelium were increased, this increased FSTL1 was derived from SGECs? Or it just colocalize with the SGECs? What about the difference between the secreted FSTL1 and intracellular FSTL1

2.     Aberrant expression of FSTL1 in SS SGs is associated with inflammatory grade, how about its association with the SG fibrosis?

3.     It is confusion about the conclusion, “Our data suggest that TGF-β1-SMAD2/3-FSTL1 regulatory circuit plays a key role in the regulation of EMT in SS”, while the author also says that “FSTL1 triggers SMAD2/3 phosphorylation in the activation of TGF-β1-dependent EMT cascade in pSS”, so FSTL1 is the downstream or the upstream of SMAD2/3 signaling?

Minor problems 

1.     Line 216-218, it seems that this is not the last version manuscript.

2.     In figure 3, the result of the control siRNA should also be shown.

3.     Some language description is confusion.

Author Response

We would like to express our sincere gratitude to the reviewer for showing interest and for the positive response to our study. All comments have been considered and the paper has been revised accordingly. All changes to the manuscript are highlighted in the text. We respond below in detail to each of the reviewer’s comments and we hope that the reviewer will find satisfactory our responses to his comments.

Major comments

  1. Follistatin-like protein 1 (FSTL1) is a secreted glycoprotein with extensive glycosylation modifications, produced primarily by cells of the mesenchymal phenotype. In this study, the data showed that FSTL1 expression in acinar cells and ductal epithelium were increased, this increased FSTL1 was derived from SGECs? Or it just colocalize with the SGECs? What about the difference between the secreted FSTL1 and intracellular FSTL1?

We thank the reviewer for his/her observation. I would like to specify that ours studies are pioneering and certainly in the next experimental projects we will also evaluate the secreted form of FSTL1 and its effects in cell cultures. Unfortunately, it is very difficult to find biopsy material to be able to set up primary cultures of healthy or diseased salivary gland cells. However, as soon as possible we will also experimentally evaluate this effect.

  1. Aberrant expression of FSTL1 in SS SGs is associated with inflammatory grade, how about its association with the SG fibrosis?

I would like to thank the reviewer for this interesting food for thought. As repeatedly highlighted in the manuscript, our studies on the role of FSTL1 are pioneering studies resulting from a recent line of investigation that has demonstrated a role of FSTL1 in the modulation of the EMT program. Although recently some researchers have identified a role of FSTL1 in the modulation of cardiac or hepatic fibrosis, a correlation between FSTl1 expression and salivary gland fibrosis has not yet been demonstrated. There is very little, but promising, evidence that FSTL1 can affect stem cell activity by intervening in regenerative processes in the salivary glands. We reserve the right to carry out a study of this kind as soon as possible on biopsy samples from SS patients using fibrotic markers.

  1. It is confusion about the conclusion, “Our data suggest that TGF-β1-SMAD2/3-FSTL1 regulatory circuit plays a key role in the regulation of EMT in SS”, while the author also says that “FSTL1 triggers SMAD2/3 phosphorylation in the activation of TGF-β1-dependent EMT cascade in pSS”, so FSTL1 is the downstream or the upstream of SMAD2/3 signaling?

We corrected the phrase reporting FSTL1 in the correct placement in the EMT activation pathway.

Minor problems 

  1. Line 216-218, it seems that this is not the last version manuscript.

We have deleted the sentence accordingly with your suggestion

  1. In figure 3, the result of the control siRNA should also be shown.

The figure 3 was modifies in concordance with your comment; a new figure 3 was created with the addition of panel B which represents healthy SGEC transfected with control scrambled siRNA. Results and Figure 3 legend were modified accordingly.

  1. Some language description is confusion.

We tried to clarify the sentences that were confusing. We hope we have satisfied your comments.

Round 2

Reviewer 2 Report

No comments